# MicroRNA-134-5p and the Extent of Arterial Occlusive Disease Are Associated with Risk of Future Adverse Cardiac and Cerebral Events in Diabetic Patients Undergoing Carotid Artery Stenting for Symptomatic Carotid Artery Disease

**DOI:** 10.3390/molecules27082472

**Published:** 2022-04-12

**Authors:** Rafał Badacz, Tadeusz Przewłocki, Piotr Pieniążek, Agnieszka Rosławiecka, Paweł Kleczyński, Jacek Legutko, Krzysztof Żmudka, Anna Kabłak-Ziembicka

**Affiliations:** 1Department of Interventional Cardiology, Institute of Cardiology, Jagiellonian University Medical College, 31-008 Krakow, Poland; rbadacz@gmail.com (R.B.); kleczu@interia.pl (P.K.); jacek.legutko@uj.edu.pl (J.L.); zmudka@icloud.com (K.Ż.); 2Clinical Department of Interventional Cardiology, John Paul II Hospital, 31-202 Krakow, Poland; tadeuszprzewlocki@op.pl (T.P.); kardio@kki.krakow.pl (P.P.); agnieszkaroslawiecka@interia.pl (A.R.); 3Department of Cardiac and Vascular Diseases, Institute of Cardiology, Jagiellonian University Medical College, 31-008 Krakow, Poland; 4Noninvasive Cardiovascular Laboratory, John Paul II Hospital, 31-202 Krakow, Poland

**Keywords:** prognostic circulating miRs, recurrent myocardial infarction and ischemic stroke, biomarkers, diabetes, carotid artery stenosis, cardiovascular events

## Abstract

There is little known about the prognostic value of serum microRNAs (miRs) in diabetic patients with symptomatic internal carotid artery disease (ICAS) who underwent stent supported angioplasty (PTA) for ICAS. The present study aimed to investigate expression levels of selected miRs for future major adverse cardiac and cerebral events (MACCE) as a marker in diabetic patients following ICAS-PTA. The expression levels of 11 chosen circulating serum miRs were compared in 37 diabetic patients with symptomatic ICAS and 64 control group patients with symptomatic ICAS, but free of diabetes. The prospective median follow-up of 84 months was performed for cardiovascular outcomes. Diabetic patients, as compared to control subjects, did not differ with respect to age (*p* = 0.159), distribution of gender (*p* = 0.375), hypertension (*p* = 0.872), hyperlipidemia (*p* = 0.203), smoking (*p* = 0.115), coronary heart disease (*p* = 0.182), lower extremities arterial disease (LEAD, *p* = 0.731), and miRs expressions except from lower miR-16-5p (*p* < 0.001). During the follow-up period, MACCE occurred in 16 (43.2%) diabetic and 26 (40.6%) non-diabetic patients (*p* = 0.624). On multivariate Cox analysis, hazard ratio (HR) and 95% Confidence Intervals (95%CI) for diabetic patients associated with MACCE were miR-134-5p (1.12; 1.05–1.21, *p* < 0.001), miR-499-5p (0.16; 0.02–1.32, *p* = 0.089), hs-CRP (1.14; 1.02–1.28; *p* = 0.022), prior myocardial infarction (8.56, 1.91–38.3, *p* = 0.004), LEAD (11.9; 2.99–47.9, *p* = 0.005), and RAS (20.2; 2.4–167.5, *p* = 0.005), while in non-diabetic subjects, only miR-16-5p (1.0006; 1.0001–1.0012, *p* = 0.016), miR-208b-3p (2.82; 0.91–8.71, *p* = 0.071), and hypertension (0.27, 0.08–0.95, *p* = 0.042) were associated with MACCE. Our study demonstrated that different circulating miRs may be prognostic for MACCE in diabetic versus non-diabetic patients with symptomatic ICAS. Higher expression levels of miR-134 were prognostic for MACCE in diabetic patients, while higher expression levels of miR-16 were prognostic in non-diabetic patients.

## 1. Introduction

Type 2 diabetes mellitus (T2DM) is a major risk factor for developing cardiovascular complications related to a progressive micro and macro angiopathy [1,2]. Hence, cardiovascular disease (CVD) and its cardiac and cerebral complications are the most prevalent causes of mortality and morbidity in diabetic populations [3,4].

Coronary heart disease is a leading cause of cardiovascular death in diabetic patients worldwide [1,3,4]. The second most frequent cause of death in diabetic patients is cerebral ischemia, either from the large or the small vessel disease with an incidence between 2.5 and 3.5 times higher in diabetic vs. non-diabetic patients [3,4]. Cerebral ischemia in diabetes is associated with high vascular dementia incidence, as well as high mortality and disability rate at short- and long-term follow-up [5]. However, limited number of tools to predict acute ischemic stroke (IS) outcome and the incidence of major cardiac and cerebral events (MACCE) in T2DM patients are available [6,7].

In all-comers populations, accumulating evidence has shown the existence of an intricate relationship between microRNAs (miRs) and the major mechanisms of IS, including energy failure, excitotoxicity, oxidative stress, inflammation, cell death, and blood–brain barrier (BBB) disruption [8,9]. For instance, middle cerebral artery occlusion was associated with the upregulation of miR-107 engaged in the excitotoxicity and with miR-126 engaged in the BBB disruption [9]. The involvement in the pathophysiology of cerebral ischemia was observed for miR-503 as an indicator of stroke severity and patients’ short-term outcome [10].

Consistently, in large vessel atherosclerotic disease, such as internal carotid artery stenosis (ICAS), several miRs have been postulated to have their role in symptom development, IS incidence, and further outcomes [11]. Among others, in patients with symptomatic ICAS, many miRs, including miR-17, miR-34a, mi-R-126, miR-133b, miR-155, miR-182 miR-208b, and miR-4909, were investigated for associations with risk of IS recurrence and MACCE incidence [12,13,14].

However, in patients with T2DM, unlike diabetes-free subjects, many differences in pathophysiology of ICAS and symptom development are observed [15].

We hypothesize that miR expression, among other factors, may differ in T2DM and diabetes-free patients with symptoms of cerebral ischemia attributed to ICAS despite performed carotid revascularization with stent supported angioplasty (PTA).

Therefore, in the present study, we aimed to compare expression levels of selected serum miRs in diabetic vs. non-diabetic patients as a potential biomarker of the outcome in patients with symptomatic ICAS referred to PTA.

## 2. Results

Compared with non-diabetic group, the T2DM group had lower expression of serum miR-16 (*p* < 0.001). There was no significant difference in age (*p* = 0.159), distribution of gender (*p* = 0.375), hypertension (*p* = 0.872), hyperlipidemia (*p* = 0.203), smoking (*p* = 0.115), prevalence of CHD (*p* = 0.182), LEAD (*p* = 0.731), levels of hs-CRP (*p* = 0.146), LDL-C (*p* = 0.292), serum creatinine (*p* = 0.361), and expression levels of the other investigated miRs between two groups, as shown in Table 1.

During the follow-up period, the rates of MACCE were similar for patients with vs. without diabetes (43.2% vs. 40.6%, *p* = 0.624). MACCE occurred in 16 T2DM patients including cardiovascular death in 11 (29.7%), non-fatal MI in 3 (8.1%), and non-fatal IS in 2 (5.4%), and in 26 non-diabetic patients, including cardiovascular death in 21 (32.8%), non-fatal MI in 1 (1.5%), and non-fatal IS in 4 (6.3%).

For T2DM patients, in univariate Cox proportional hazard analysis, MACCE risk was associated with a higher expression level of miR 134-5p (*p* = 0.02), miR-16-5p (*p* = 0.048), and miR-499-5p (*p* = 0.04). There was a trend towards statistical significance for miR-133a-3p (*p* = 0.089), miR-208b-3p (*p* = 0.071) and miR-34a-5p (*p* = 0.07), while there were no significant associations with the other studied miRs (Table 2). There was a significant association between MACCE incidence and prior MI (*p* = 0.029), LEAD (*p* = 0.004), and RAS (*p* = 0.028), but not with traditional cardiovascular risk factors (Table 2). Among the biochemical results, the association was found for hs-CRP (*p* = 0.025) and a trend to significance for creatinine level (*p* = 0.056).

For diabetes-free patients, MACCE risk was associated with higher expression levels of miR-1-3p (*p* = 0.022) and miR-16-5p (*p* = 0.019) Table 2. There was a trend to significance between MACCE incidence and arterial hypertension (*p* = 0.067) and fibrinogen level (*p* = 0.070), but not with the other cardiovascular risk factors. We found a trend of association between MACCE and LEAD (*p* = 0.081).

In the multivariate Cox analysis, HRs and 95% CIs for T2DM patients associated with MACCE were as follows: miR-134-5p (1.12; 1.05–1.21, *p* < 0.001), miR-499-5p (0.16; 0.02–1.32, *p* = 0.089), hs-CRP (1.14; 1.02–1.28; *p* = 0.022), prior MI (8.56, 1.91–38.3, *p* = 0.004), LEAD (11.9; 2.99–47.9, *p* = 0.005), and RAS (20.2; 2.4–167.5, *p* = 0.005).

In non-diabetic subjects, only miR-16-5p (1.0006; 1.0001–1.0012, *p* = 0.016), miR-208b-3p (2.82; 0.91–8.71, *p* = 0.071), and hypertension (0.27, 0.08–0.95, *p* = 0.042) were associated with MACCE. The detailed parameters of multivariate Cox hazard analysis are shown in Table 3.

## 3. Discussion

Patients with T2DM, unlike diabetes-free subjects, suffer more often from cardiovascular events [3,16]. Between the years 2007 and 2017, CVD was the cause of death in 9.9% of T2DM patients (representing 50.3% of all deaths) [3]. Despite improvements in cardiac care, T2DM still increases the risk of death, including all-cause (1.68; 95%CI 1.60 to 1.78), CVD (1.61; 95%CI 1.47 to 1.76), and MI (1.59; 95%CI 1.27 to 1.99) [16]. In the TECOS Trial, out of 530 cardiovascular deaths, sudden death accounted for 27.3% fatality cases, followed by stroke (12.3%), heart failure (12%), and MI (9%) [4].

Only few recent studies addressed the possible role of miRs modulation in patients with both symptomatic and asymptomatic ICAS and concomitant T2DM [17,18].

The novelty of the present study is the implication of the different diagnostic and prognostic circulating miRs associated with cerebral ischemia resulting from symptomatic ICAS in diabetic vs. non-diabetic patients.

We identified one miR (miR-16) that differed among T2DM vs. non-diabetic patients. We found higher miR-16 expression levels in patients without diabetes. Higher miR-16 expression was also identified as prognostic miRs of future MACCE, but only in diabetes-free patients.

As previously postulated, carotid or peripheral ischemia can negatively influence remote vascular remodeling [19,20]. The possible mechanism of miR-16 negative action can be through the injury promotion in a remote vascular district [19,20]. For example, in a study by Sorrentino et al., miR-16 was upregulated after vascular injury in a rat model in the presence of limb ischemia [21]. This was associated with a negative effect on endothelial repair reducing nitric oxide bioavailability [21]. Thus, limb ischemia affected negative carotid remodeling increasing neo-intima formation and delayed re-endothelialization after the injury [21]. Additionally, as previously shown, miR-16 can be associated with specific plaque features, such as plaque ulceration or calcification [22].

MiR-16-5p was also identified as upregulated in MI or coronary artery disease patients [23]. In another study, the acute MI patients in above the median levels of plasma miR-16 group suffered a 1.87-fold higher risk of MI recurrence compared to patients with a lower median value (*p* = 0.029) [24]. This is in line with our previous findings, where expression levels of miR-1-3p, miR-16-5p, and miR-122-5p were independent risk factors of MACCE [25].

We have observed two miRs to be prognostic for MACCE in T2DM patients, but not in the diabetes-free patients; for example, miR-134, which was associated with a 1.12-fold risk increase for MACCE, and miR-499, which showed a negative relationship with risk of future MACCE. MiR-134 after birth is restricted to the brain [26], and regulates ischemia/reperfusion injury-mediated neuronal cell death by targeting heat shock protein A12B (HSPA12B) and cyclic AMP response element-binding protein (CREB) [27,28]. There is a relationship between miR-134 and chronic inflammation, represented by hs-CRP and tumor necrosis factor alpha (TNF-α) [29]. In our present study, we have found elevated levels of hs-CRP to be associated with a 1.14-fold risk increase in MACCE in T2DM. Importantly, TNF-α, a cytokine produced in the adipose tissue, is associated with insulin resistance [30]. Induction of miR-134-5p by TNF-α has been observed and suggests a potential role for miR-134-5p in insulin-mediated glucose disposal and insulin sensitivity [29,30]. Furthermore, miR-134 plays role in developing diabetic nephropathy [31].

Altogether, a chronic low-grade inflammation, insulin-resistance, and diabetic nephropathy are risk factors for cardiovascular events [32]. However, to what extent miR-134 expression during cerebral ischemia exerts its impact on the future MACCE risk in diabetic patients and which mechanism plays a crucial role needs future investigations.

There is a link between miR-134 expression levels and stroke recurrence and MI incidence. A recent miR microarray analysis revealed that the expression of miR-19b, miR-134, and miR-186 were upregulated in patients with acute coronary syndrome compared to controls [33]. Interestingly, recent studies indicated that miR-134 might be used as a potential biomarker of coronary artery calcification, unstable coronary artery disease, or MI [34]. In line, carotid echolucent (unstable) plaques as compared to echogenic plaques differed in levels of miRs, including higher expression levels of miR-134-5p (*p* = 0.042) [22].

On the other hand, a study by Pielok et al. proved that miR-499 may be engaged in the hepatic insulin resistance and the development of metabolic diseases [35]. However, the clinical value of miR-499 for assessment of the MACCE incidence remains unclear.

The idea of ‘the otherness’ in miRs mechanisms for the outcome was also postulated in patients with various cardiovascular risk factors, e.g., in smokers in the context of the LEAD [36].

Our present study demonstrated a particularly huge role of the co-coexisting atherosclerotic lesions across the other vascular arterial beds in patients with T2DM. LEAD, coronary artery disease with a prior MI, and RAS were associated with 11.9-fold, 8.56-fold, and 20.2-fold higher MACCE incidence, respectively, in T2DM vs. non-diabetic patients. As previously reported, multiterritory atherosclerotic disease is frequent in diabetic patients [37,38].

Furthermore, arterial vasculopathy pattern differs substantially from that seen in patients with atherosclerotic disease but free from diabetes. Atherosclerotic lesions in diabetic coronary and peripheral arteries are much more disseminated and complex, leading to smaller vessels diameters, and as a consequence, suboptimal interventional treatment results [39]. Silent cardiac ischemia is more prevalent in T2DM vs. non-diabetic patients, accounting for 10–20% vs. 1–4%, respectively [40,41].

A major challenge associated with diabetes management for the reduction in cardiovascular events is the complex and multifaceted nature of the relationship linking diabetes to CVD [40]. Apart from traditional risk factors, such as age, male gender, dyslipidemia, hypertension, and smoking, in patients with T2DM, there are many non-traditional cardiovascular risk factors that elevate a person with diabetes to a higher risk category [41,42].

The diabetes-specific cardiovascular risk factors of atherosclerosis progression are the body fat distribution, metabolic syndrome, subclinical chronic inflammation, insulin resistance, increased glycosylation and oxidation, and disturbances in glucose metabolism [43]. Additionally, hyperinsulinemia and insulin resistance are associated with an increased free fatty-acid release promoting high triglycerides levels, high levels of ApoB and VLDL, and a low high-density lipoprotein (HDL) cholesterol level [43].

There is also otherness between specific circulating biomarkers and cytokines with the course of diabetes [43,44]. In diabetic patients, there is a greater role of inflammation for developing diabetes complications, such as hs-CRP, Interleukin-1 and -6, TNF-α, and expression of specific miRs, belonging to a family of small non-coding RNAs [18]. We observed the prognostic role of miR-134 and lower expression levels of miR-16 in our patients with diabetes and symptomatic ICAS.

Furthermore, diabetes exacerbates the proliferative phenotype of vascular smooth muscle cells (VSMCs), which underlies the very high rate of vascular complications in patients with diabetes. There is evidence that the contemporary miR-29c overexpression and miR-204 inhibition in the injured artery reduced coronary arterial stenosis in diabetic rats by preventing the exaggerated VSMCs growth upon injury [45]. In line with this, miR-503-5p might be a potential diagnostic biomarker for asymptomatic ICAS and overexpression of miR-503-5p may inhibit the proliferation of VSMCs and reduce stenosis severity [46].

Early detection of biomarkers associated with future outcome in patients with T2DM to prevent various cardiovascular events, such as MI, IS, and death, is of great interest. In the present study, we demonstrated higher expression levels of miR-134, a microRNA specific for cerebral ischemia injury, that showed prognostic role for future adverse cardiovascular events in T2DM. Moreover, patients with symptomatic ICAS and T2DM presented with lower expression levels of miR-16. Although high expression of miR-16 was prognostic for future MACCE in non-diabetic patients, the prognostic value of mentioned miR was not proven for patients with diabetes.

## 4. Materials and Methods

### 4.1. Study Population

In this prospective study, we evaluated 101 patients with hemodynamically significant symptomatic ICAS admitted to Endovascular and Vascular Surgery Department at our institution with the aim of PTA between January 2013 and January 2014. In all patients, revascularization of the target ICAS was performed by carotid artery stenting, followed by a 7-year follow-up period.

The study group comprised 37 patients with symptomatic ICAS causing 50–99% stenosis and concomitant T2DM. The control group consisted of 64 diabetes-free patients with symptomatic ICAS. Both diabetic and non-diabetic patients underwent PTA for symptomatic lesion according to the guidelines [47,48].

Inclusion criteria were: ICAS exceeding 50% lumen reduction confirmed by imaging studies (Doppler ultrasound or Computed tomography angiography (angio-CT) in the territory of cerebral ischemia with relevant brain imaging findings and/or neurological symptoms, confirming the association of carotid stenosis with cerebral ischemia as ensured by the consulting neurologist.

Exclusion criteria for both study and control groups included: acute heart failure or congestive heart failure in class III and IV of New York Heart Association (NYHA) classification, acute coronary syndrome, no direct association between ICAS and neurological symptoms or lesions on brain CT tomography, any active neoplastic disease, chronic or acute systemic inflammatory condition, and any known or suspected active infection.

The distribution of traditional cardiovascular risk factors (hyperlipidemia, arterial hypertension, former or active smoking), as well as the history of coronary artery disease including history of myocardial infarction (MI), renal artery stenosis (RAS), and peripheral extremities arterial disease (LEAD) were recorded. Definitions of the above were adopted from the scientific statements of the European Society of Cardiology [49,50,51]. The data on participants’ comorbidities were acquired from available medical history or based upon presented symptoms supported by diagnostic tools, e.g., Doppler ultrasonography, computed tomography or magnetic resonance imaging, and eventually angiography. All patients obtained peri- and post-procedural optimal medical treatment according to recommendations of respective societies [49,50,51].

The study complies with the Declaration of Helsinki and was approved by the Jagiellonian University Ethics Committee (KBET/21/B/2012; date of approval: 25 October 2012 with further extensions). All participants signed a written informed consent.

### 4.2. Biochemical Tests and miRs Extraction

All patients had fasting blood samples obtained on patient admission to the department, prior to PTA procedure, as soon as the signed informed consent was obtained. Serum blood tests included high-sensitivity C-reactive protein (hs-CRP), fibrinogen, creatinine, and low-density-lipoprotein (LDL) cholesterol levels.

Peripheral blood serum samples for profiling miRs were collected on patient admission before heparin treatment. Samples were left to coagulate for 30 min and centrifuged, and sera were frozen at –80 °C until analysis.

We used the miRNeasy Serum/Plasma Kit (cat. No. 217184, Qiagen, Hilden, Germany) with the beginning lysis by Trizol LS Reagent (cat. No. 10296-028, Invitrogen, Waltham, MA, USA) for the extraction of miRs. The RNA yield and concentrations were determined by capillary electrophoresis on the Agilent Bioanalyser 2100 with the Eukaryote Total RNA Pico Chip (Agilent Technologies, Inc, Santa Clara, CA, USA). An average of 60 ± 31.9 pg/μL total RNA from 300 μL of serum was collected.

The following sequence and catalog numbers for circulating miRs were used in each case: miR-1-3p (UGGAAUGUAAAGAAGUAUGUAU; EQ-204344), miR-34a-5p (UGGCAGUGUCUUAGCUGGUUGU; EQ-204486), miR-122-5p (UGGAGUGUGACAAUGGUGUUUG; EQ-205664), miR-124-3p (UUUGGUCCCCUUCAACCAGCUG; EQ-204788), miR-133a-3p (UUUGGUCCCCUUCAACCAGCUG; EQ-204788), miR-133b (UUUGGUCCCCUUCAACCAGCUA; EQ-204162), miR-134-5p (UGUGACUGGUUGACCAGAGGGG; EQ-205896), miR-208b-3p (AUAAGACGAACAAAAGGUUUGU; EQ-204636), miR-375 (UUUGUUCGUUCGGCUCGCGUGA; EQ-204362), and miR-499-5p (UUAAGACUUGCAGUGAUGUUU; EQ-205935). The endogenous miR-16-5p (UAGCAGCACGUAAAUAUUGGCG; EQ-204409) was used as a reference.

Analyzed miRs were taken into consideration based on the data regarding their potential relationship with development of atherosclerosis (PubMed, Bethesda, MD, USA), and their potential prognostic value.

At the time of the study, Exiqon LNA primers were used to quantify 10 mature miRs using the ViiA 7 real-time PCR system equipped with a 384-well reaction plate (Life Technologies, Carlsbad, CA, USA). RNA was converted to cDNA using the Universal cDNA Synthesis Kit (cat. No. EQ-203300, Exiqon, Vedbæk, Denmark). Before synthesis, RNAs were spiked with a synthetic miRNA that served as a control for the cDNA synthesis reaction. Real-time PCR was performed in triplicate with SYBR Green master mix Universal RT (cat. No. EQ-203400, Exiqon, Vedbæk, Denmark) using standard conditions.

Data were processed by the delta-Ct method, using a global normalization approach as implemented in the open source DataAssist software (Life Technologies, Carlsbad, CA, USA). The fold changes (RQ) were calculated, and statistically significant variations between group samples were filtered by the calculation of adjusted *p*-values using the Benjamini–Hochberg false discovery rate.

### 4.3. Carotid Artery Stenting Procedure

Between January 2013 and January 2014, 101 carotid artery stenting procedures were performed for symptomatic ICAS, exceeding at least 50% lumen reduction, according to the ‘tailored-CAS’ algorithm [52,53].

### 4.4. Follow-Up and Reporting of MACCE

The incidences of cardiovascular death, MI, and IS as well as composite endpoint (MACCE) were recorded prospectively during a follow-up period of 7 years. Adverse events were defined as fatal or non-fatal IS, fatal, or non-fatal MI or cardiovascular death (i.e., any sudden or unexpected death unless proven as non-cardiovascular on autopsy). MI was diagnosed according to criteria of the European Society of Cardiology. Diagnosis of IS was to be given by a neurologist to ensure reliability.

Final visits were done through telephone contact with a patient or appointed family member. One patient was lost to follow-up; however, the data on patient vital status were obtained from the national health registry.

### 4.5. Statistical Analysis

Continuous variables are presented as mean ± one standard deviation (SD) for variables with proven normal distribution by Shapiro–Wilk test, and median with interquartile range (IQR) for variables with no normal distribution. Categorical variables are expressed as frequencies and percentages (*n*, %). Means of analyzed parameters across groups were tested with the analysis of variance (ANOVA) test, and frequencies were compared by the chi-square test for independence. The potential independent prognostic markers of MACCE during the follow-up period were established from the clinical, biochemical, and miR variables with a Cox proportional hazard univariate analysis, and in case of a trend toward difference (*p* < 0.1), they were entered into a multivariate Cox proportional hazard analysis model. The results of uni- and multivariate Cox proportional hazard analysis were expressed as hazard ratio (HR) and 95% confidence interval (95%CI). Statistical analyses were performed with Statistica 13.0 software. Statistical significance was assumed at a *p*-value < 0.05.

## 5. Conclusions

Our study demonstrated that different circulating miRs may be prognostic for MACCE in diabetic versus non-diabetic patients with symptomatic ICAS. Higher expression levels of miR-134 were prognostic for MACCE in diabetic patients, while higher expression levels of miR-16 were prognostic for MACCE in non-diabetic patients.

## 6. Study Limitations

Our study has some limitations. The results are acquired form a single-center study with a relatively small group. The patients scheduled for surgical carotid endarterectomy were not enrolled in the present study. These factors could have an impact on the lower study power (0.762). Thus, future studies, preferably multicenter, are needed to assess the specific miRs as predictors of cardiovascular outcomes in patients with T2MD and symptomatic ICAS.

## Figures and Tables

**Table 1 molecules-27-02472-t001:** Baseline patients’ characteristics. Comparison of patients with symptomatic internal carotid artery stenosis with diabetes and without diabetes.

Parameter	All *n* = 101	Diabetic *n* = 37	Non-Diabetic *n* = 64	*p*-Value
Demographic data				
Age, (median; IQR)	69; 62–76	71; 63–78	67.5; 61.5–74	0.159
Male gender, *n* (%)	63 (62.3%)	21 (56.8%)	42 (65.6%)	0.375
Hypertension, *n* (%)	96 (95.0%)	35 (94.5%)	61 (95.3%)	0.872
Hypercholesterolemia, *n* (%)	87 (86.1%)	34 (91.8%)	53 (82.8%)	0.203
Smoking habit, *n* (%)	62 (61.3%)	19 (51.3%)	43 (67.2%)	0.115
Coronary artery disease, *n* (%) *	54 (53.4%)	23 (62.2%)	31 (48.4%)	0.182
Lower extremities arterial disease, *n* (%) *	28 (27.7%)	11 (29.7%)	17 (26.6%)	0.731
Prior myocardial infarction, *n* (%)	20 (19.8%)	8 (21.6%)	12 (18.7%)	0.727
Renal artery stenosis, *n* (%) *	7 (6.9%)	2 (5.4%)	5 (7.8%)	0.646
Laboratory results (serum)				
Serum creatinine, μmol/L, (median; IQR)	82; 70–100	85; 71–101	81; 68.5–99	0.361
C-Reactive Protein, g/L, (median; IQR)	2.59; 1.99–250	3.15; 1.83–6.29	2.21; 1.27–4.45	0.146
Fibrinogen, g/L, (median; IQR)	3.51; 3.01–4.30	3.78; 3.33–4.62	3.40; 3.00–4.00	0.062
LDL-cholesterol, mmol/L, (median; IQR)	2.65; 1.99–3.04	2.59; 1.94–3.46	2.56; 2.03–2.95	0.292
microRNA				
miR-1-3p, A.U., (median; IQR)	0.17; 0.08–0.32	0.15; 0.08–0.24	0.19; 0.08–0.38	0.227
miR-122-5p, A.U., (median; IQR)	48.05; 19.43–250.4	40.95; 12.72–142.4	52.82; 28.63–310.4	0.146
miR-124-3p, A.U., (median; IQR)	0.23; 0.09–0.63	0.24; 0.09–0.57	0.22; 0.08–0.66	0.625
miR-133a-3p, A.U., (median; IQR)	0.87; 0.63–1.22	0.85; 0.62–1.26	0.87; 0.63–1.15	0.805
miR-133b, A.U., (median; IQR)	1.87; 1.19–2.53	1.87; 1.15–2.60	1.69; 1.24–2.47	0.766
miR-134-5p, A.U., (median; IQR)	0.82; 0.33–2.80	0.90; 0.43–3.17	0.73; 0.29–1.87	0.357
miR-16-5p, A.U., (median; IQR)	94.57; 37.57–263.7	45.32; 14.60–71.19	122.78; 64.96–543	<0.001
miR-208b-3p, A.U., (median; IQR)	0.005; 0.002–0.022	0.005; 0.002–0.018	0.005; 0.002–0.02	0.978
miR-34a-5p, A.U., (median; IQR)	0.72; 0.31–1.06	0.76; 0.48–1.11	0.63; 0.28–1.04	0.145
miR-375, A.U., (median; IQR)	3.53; 1.62–10.21	3.11; 1.49–6.71	4.48; 1.69–21.15	0.226
miR-499-5p, A.U., (median; IQR)	0.02; 0.01–0.04	0.02; 0.01–0.07	0.02; 0.01–0.04	0.605

A.U., arbitrary units; *—defined as the presence of arterial stenosis exceeding 50% lumen reduction on angiography.

**Table 2 molecules-27-02472-t002:** Univariate Cox analysis for the incidence of major cardiac and cerebral events (MACCE) in diabetic and non-diabetic patients.

Prognostic Factors	Diabetic Patients		Non-Diabetic Patients	
	**HR (95% CI)**	***p*-Value**	**HR (95% CI)**	***p*-Value**
microRNA				
miR-1-3p	3.93 (0.57–27.06)	0.164	3.72 (1.21–11.5)	0.022
miR-122-5p	1.01 (0.99–1.02)	0.678	1.00 (0.99–1.01)	0.412
miR-124-3p	0.81 (0.40–1.64)	0.568	1.34 (0.76–2.38)	0.305
miR-133a-3p	2.13 (0.88–5.11)	0.089	0.89 (0.60–1.32)	0.573
miR-133b	0.87 (0.53–1.42)	0.581	0.98 (0.80–1.20)	0.862
miR-134-5p	1.04 (1.01–1.07)	0.020	1.05 (0.98–1.13)	0.140
miR-16-5p	1.01 (1.00–1.02)	0.048	1.0006 (1.0001–1.001)	0.019
miR-208b-3p	4.42 (0.87–22.25)	0.071	2.77 (0.84–9.15)	0.095
miR-34a-5p	0.41 (0.15–1.07)	0.070	1.04 (0.80–1.36)	0.749
miR-375	1.02 (0.98–1.06)	0.325	1.01 (0.99–1.02)	0.856
miR-499-5p	4.84 (1.07–21.89)	0.040	0.36 (0.02–11.13)	0.566
Demographic data				
Age	1.02 (0.96–1.08)	0.457	1.02 (0.97–1.06)	0.375
Male gender	0.57 (0.19–1.69)	0.317	0.84 (0.36–1.97)	0.700
Hypertension	n.a.	n.a.	0.32 (0.09–1.08)	0.067
Hiperlipidemia	1.56 (0.20–11.89)	0.667	1.07 (0.37–3.13)	0.897
Smoking habit	1.12 (0.41–3.13)	0.817	1.69 (0.31–2.53)	0.369
CAD	1.38 (0.46–4.08)	0.558	0.97 (0.44–2.14)	0.956
LEAD	4.41 (1.57–12.42)	0.004	2.04 (0.91–4.55)	0.081
Prior MI	3.39 (1.13–10.20)	0.029	1.05 (0.39–2.82)	0.921
Renal artery stenosis	5.72 (1.20–27.21)	0.028	1.25 (0.49–3.15)	0.635
Laboratory results				
Serum creatinine	1.02 (0.99–1.01)	0.056	1.00 (0.99–1.01)	0.646
C-Reactive Protein	1.10 (1.01–1.20)	0.025	0.96 (0.90–1.04)	0.392
Fibrinogen	1.25 (0.82–1.90)	0.309	0.60 (0.34–1.04)	0.070
LDL-cholesterol	0.90 (0.61–1.35)	0.617	1.37 (0.83–2.29)	0.221

CAD, Coronary Artery Disease; CI, Confidence Interval; HR, Hazard Ratio; LDL, Low Density Lipoprotein; LEAD, Lower Extremities Arterial Disease; MI, Myocardial Infarction.

**Table 3 molecules-27-02472-t003:** Multivariate Cox analysis for cardiovascular outcome in diabetic and non-diabetic patients for parameters that were significant in the Univariate Cox analysis.

Study Group	Prognostic Factors	HR (95% CI)	*p*-Value
Patients with diabetes	miR-134-5p	1.12 (1.05–1.21)	0.028
	hs-CRP	1.14 (1.01–1.28)	0.022
	prior MI	8.56 (1.91–38.25)	0.004
	LEAD	11.98 (2.99–48.0)	<0.001
	RAS	20.24 (2.44–167.5)	0.005
	miR-499-5p	0.16 (0.02–1.32)	0.089
	miR-133a-3p	2.12 (0.51–8.91)	0.302
	miR-16-5p	1.01 (0.99–1.02)	0.410
	miR-208b-3p	5.91 (0.01–7.93)	0.314
	miR 34a-5p	0.65 (0.15–2.67)	0.552
Patients diabetes-free	miR-16-5p	1.0006 (1.0001–1.0011)	0.016
	hypertension	0.27 (0.07–0.95)	0.042
	miR-208b-3p	2.82 (0.91–8.71)	0.071
	miR-1-3p	0.58 (0.06–5.22)	0.628
	Fibrinogen	0.31 (0.08–1.12)	0.171

## Data Availability

The data presented in this study are available on request from the corresponding author. The data are not publicly available due to privacy.

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
