# Peer review of "MicroRNA-134-5p and the Extent of Arterial Occlusive Disease Are Associated with Risk of Future Adverse Cardiac and Cerebral Events in Diabetic Patients Undergoing Carotid Artery Stenting for Symptomatic Carotid Artery Disease"

_molecules, 2022, doi:10.3390/molecules27082472_

Round 1

Reviewer 1 Report

Authors have nice and interesting hypothesis. If micro mRNA levels differ between CAS patients with and without diabetes and if these differences would further have value in risk analyses for later MACCE.

The major concern is the sample size. Cohort size is 101 patients and only 37 were diabetic (16 primary outcomes). The power analyses should be presented for anticipated difference and discussed if anticipated sample size was not reached. Also, it would be nice if manuscript would be more carefully prepared. Already the early words of introduction seem to need some revision. This kind of errors do not support the confidence of reviewer of reader. Also, word artero-occlusive is a little unknown and would possibly refer to different arterial status than CAS patients should have?

Abstract: materials and methods might be more in general and not a list of mRNAs? Results focus on main observations? Conclusions do the results really support this?

Introduction: Lines 74-79 should be revised to be more clear (reader friendly)

Materials and methods.

Study population: a more detailed description would be nice. At one point it is stated that patients underwent stenting and on other only PTA, institute/institutes participating patient collection…..

Lines 101-107 might be nice to be more accurate, LEAD really based on ABI measurements….

At least at some point state that open surgery of ICAS was not available at this institute or any of the patients were treated with open surgery.

The follow-up protocol should be more detailed.

Results: The text on results section could be shorter since many of the written results are shown on tables 1 and 2.   Table 1 might be more visual with revision? Table 2; The table would benefit from revision. Now it required some thinking to understand grouping. All 101 patients (study group). Label should then be one line lower in the table? Or first diabetic patients (37) divided with line, not logically placed (in relation to the label diabetes free) later in the table….

Discussion and conclusions:

Please see above. The power analyses for the study should be presented since the results are intensively discussed.

I also think the inclusion criteria is usually looser (p<0.2-0.1) in the univariate analyses when selecting variables for multivariable analyses?

Data availability statement is missing?

Reviewer 2 Report

In the present study authors demonstrated higher expression levels of miR-134, a microRNA specific for cerebral ischemia injury, that showed prognostic role for future adverse cardiovascular events in diabetic patients. Patients with symptomatic  ICAS and diabetes presented with lower expression levels of miR-16. Although high expression of miR-16 was prognostic for future MACCE in non-diabetic patients, the prognostic value of miR was not proven for patients with diabetes.

Abstract: Conclusion in abstract should be in accordance with Conclusion remarks in the body of the text.

Title of Introduction:

  1. There is something wrong with the Introduction first line: “Intro prognostic circulating miRs; recurrent myocardial infarction and ischemic 43 stroke; biomarkers; diabetes; carotid artery stenosis; cardiovascular eventsduction”
  2. otherness in diabetes should be clarified

Design

The power of the study should be notified.

Results

The univariate analysis is difficult for understanding and two tables should be added for the following text.

“For diabetic patients, in univariate analysis, increasing MACCE risk was associated 204 with a higher expression levels of miR 134-5p (HR 1.04, 95%CI 1.01 to 1.07, P = 0.02), miR- 205 16-5p (HR 1.01, 95%CI 1.00 to 1.02, P = 0.048) and miR-499-5p (HR 4.84, 95%CI 1.07 to 206 21.89, P = 0.04), while there was no significant association for miR-1-3p (P = 0.164), miR- 207 122-5p (P = 0.678), miR-124-3p (P = 0.568); miR-133b (P = 0.581) or miR-375 (P = 0.325). 208 There was a trend towards statistical significance for miR-133a-3p (HR 2.13, 95%CI 0.88 to 209 5.11, P = 0.089), miR-208b-3p (HR 4.42, 95%CI 0.87 to 22.25, P = 0.071) and miR-34a-5p (HR 210 0.41, 95%CI 0.15 to 1.07, P = 0.07). There was no significant association between MACCE 211 and traditional cardiovascular risk factors like hyperlipidemia (P = 0.667), hypertension 212 (P = 0.994), smoking (P = 0.817), male gender (0.317), or age (0.457). However, there was a 213 significant association between MACCE incidence and prior MI (HR 3.39, 95%CI 1.13 to 214 10.20, P = 0.029), LEAD (HR 4.41, 95%CI 1.57 to 12.42, P = 0.004), as well as RAS (HR 5.72, 215 95%CI 1.20 to 27.21, P = 0.028). Among laboratory results, the correlation was confirmed 216 for hs-CRP (HR 1.10, 95%CI 1.01 to 1.20, P = 0.025) and a trend to significance for creatinine 217 (HR 1.02, 95%CI 0.99 to 1.01, P = 0.056).

For diabetes-free patients, in univariate analysis, increasing MACCE risk was associ- 219 ated with higher expression levels of miR-1-3p (HR 3.72, 95%CI 1.21 to 11.5, P = 0.022), 220 miR-16-5p (HR 1.0006, 95%CI 1.0001 to 1.001, P = 0.019), miR-208b-3p (HR 2.77, 95%CI 221 0.84 to 9.15, P = 0.095), but not with miR-122-5p (P = 0.412), miR-124-3p (P = 0.305), miR- 222 133a-3p (P = 0.573), miR-133b-3p (P = 0.862), miR-134a-5p (P = 0.140), miR-34a-5p (P = 223 0.749), miR-375 (P = 0.856), and miR-499-5p (P = 0.566). There was trend to significant as- 224 sociation between MACCE incidence and arterial hypertension (P = 0.067) and fibrinogen 225 level (P = 0.070), but not with the other cardiovascular risk factors like hyperlipidemia (P 226 = 0.897), LDL-C level (P = 0.221), age (P = 0.375), smoking (P = 0.369), male gender (P = 227 0.700), or hs-CRP level (P = 0.392) and creatinine (P = 0.646). Also, there was a trend to 228 association between risk of MACCE and LEAD (HR 2.04, 95%CI 0.91 to 4.55, P = 0.081), 229 but not CHD (P = 0.956). “

Round 2

Reviewer 1 Report

Authors have revised the manuscript according presented comments. I have no further comments.

Reviewer 2 Report

The authors have addressed all comments.